# Effects of Walking Speed and Added Mass on Hip Joint Quasi-Stiffness in Healthy Young and Middle-Aged Adults

**DOI:** 10.3390/s23094517

**Published:** 2023-05-06

**Authors:** Shanpu Fang, Vinayak Vijayan, Megan E. Reissman, Allison L. Kinney, Timothy Reissman

**Affiliations:** Department of Mechanical and Aerospace Engineering, University of Dayton, Dayton, OH 45469, USA

**Keywords:** joint dynamics, load carriage, biomechanics, exoskeletons, rehabilitation

## Abstract

Joint quasi-stiffness has been often used to inform exoskeleton design. Further understanding of hip quasi-stiffness is needed to design hip exoskeletons. Of interest are wearer responses to walking speed changes with added mass of the exoskeleton. This study analyzed hip quasi-stiffness at 3 walking speed levels and 9 added mass distributions among 13 young and 16 middle-aged adults during mid-stance hip extension and late-stance hip flexion. Compared to young adults, middle-aged adults maintained a higher quasi-stiffness with a smaller range. For a faster walking speed, both age groups increased extension and flexion quasi-stiffness. With mass evenly distributed on the pelvis and thighs or biased to the pelvis, both groups maintained or increased extension quasi-stiffness. With mass biased to the thighs, middle-aged adults maintained or decreased extension quasi-stiffness while young adults increased it. Young adults decreased flexion quasi-stiffness with added mass but not in any generalizable pattern with mass amounts or distributions. Conversely, middle-aged adults maintained or decreased flexion quasi-stiffness with even distribution on the pelvis and thighs or biased to the pelvis, while no change occurred if biased to the thighs. In conclusion, these results can guide the design of a hip exoskeleton’s size and mass distribution according to the intended user’s age.

## 1. Introduction

Joint quasi-stiffness [1], referred to by some as joint stiffness [2], represents the relationship between a person’s internal moment and angular change for a certain plane of a joint. This relationship not only provides a characterization of the joint mechanical properties but also represents a lower-order model approximation of the neural mechanisms of human motor control [3]. In particular, researchers have made efforts in determining how joint quasi-stiffness during gait differs by gender [4], age [2,4,5], body dimensions [6,7,8], walking speeds [2,5], ground surface conditions [9], and body loads [10,11]. However, it should be mentioned that quasi-stiffness has been more characterized in the literature at the ankle joint [2,4,5,8,10,11] and knee joint [2,7,9,10] than at the hip joint [2,6,10].

This basic science understanding of joint quasi-stiffness has also been applied to help make clinical decisions [12,13], describe dynamic joint function [4], and advance hardware designs of wearable assistive devices [5,9]. With respect to the latter, recent advances in assistive devices known as exoskeletons (exos) often use this knowledge of joint quasi-stiffness to determine control methods which can cooperatively interact with the user in order to provide assistance for desired movements [14,15]. With that being said, achieving assistance can be challenging, as all exos introduce some changes to the wearer’s biomechanics via the influence of the device’s added mass magnitudes and body locations [16,17,18,19]. As hip exos add the device’s mass closer to the wearer’s center of body mass, there is an assumption that hip exos will have reduced influence on the wearer’s biomechanics and, therefore, hip exos have recently gained interest [20]. However, this added mass distribution effect on human motor control, or joint quasi-stiffness, has not received enough attention to make such an assumption. In a review of the literature, if mass amounts are exceeded at specific limb locations, even for those of hip exo designs, then wearer adaptations can result in significant changes in their hip angular and moment profiles, which are related to hip joint quasi-stiffness [17,18,21,22]. Specifically, among healthy young adults, walking with as little as 10.8 lb added mass around the pelvis increased hip flexion angles during the majority of the gait cycle and hip extension moment during the first half of the gait cycle [17,18]. In addition, walking with a backpack or vest [21,23], or with weights around the pelvis and/or thighs bilaterally [18,19,24], also changed other lower-limb joint angular and/or moment profiles. Thus, hip exo physical designs, and their controllers, can benefit from incorporating knowledge of such responses by the wearer with respect to quasi-stiffness at the hip joint during walking. A reasonable starting point is with understanding the actual influence of wearing added mass that is representative of hip exos and then extending this to increased walking speeds, as that is a common goal of exo assistance [20,25].

While all exos have the intention of providing assistance, a hip exo achieves this by altering the resultant torque at the wearer’s hip joint. Thus, depending on the design, the change in the resultant torque at the hip joint can lower or increase the stiffness felt by the hip exo wearer. Both passive and active hip exo designs can benefit from knowing the response of the wearer, specifically regarding their hip joint quasi-stiffness at different phases of the gait cycle [26,27,28,29,30,31,32,33]. For instance, passive hip exos, with components such as springs, can be designed to be lightweight and therefore are thought to minimize the added mass effect on biomechanics. However, overall they do not change the net mechanical energy at the actuated joints, but instead provide assistance by selectively storing and returning mechanical energy only at specified times of the gait cycle [20]. Thus, mechanical tuning is critical to the success of passive hip exos. Parallel research has focused on active exos, which have the primary advantage of being able to change the net mechanical energy at the actuated joints during their wearer’s movement. These devices tend to be heavier and thus more likely to change the wearer’s biomechanics due to added mass [20]. However, their potential for tailoring the assistance of varying profiles has made them quite versatile to tuning for assistance. With respect to developing their control algorithms, a popular one is assist-as-needed (AAN) [15,34,35,36], where the exo provides only necessary, or intermittent, assistance to the wearer in order to achieve predefined gait profiles. Thus, the wearer remains in complete control of their movement when not active and must adapt accordingly to the presence of wearing the device. Another popular algorithm is impedance control [37], which alters the total resistance of the exo and the wearer’s joint by varying the assistance to match the desired model parameters of inertia, damping, and stiffness. For both algorithms and many others like them, estimates of what profiles the wearer would naturally impose for their movement is critical to their tuning. For example, two active hip exos [38,39] presented in a recent exo review paper [25] exerted net mechanical energy only during hip flexion and/or extension phases of the gait cycle. To further tune such active control algorithms or even passive hip exo designs, knowledge of the quasi-stiffness during these phases would allow for better cooperative control. Specifically, the quasi-stiffness in the mid-to-late-stance phase, when a person exerts higher internal hip joint moments near the maximum hip extension angle, could be incorporated into the tuning parameters, and likewise the timing relationship during the pre-swing phase after that [40].

To date, aspects of the relationships between hip joint quasi-stiffness and walking speed, body mass, and body height have all been quantified for the hip extension and flexion phases of the gait cycle [6]. In particular, walking at an increased speed was found to increase hip joint quasi-stiffness during the early stance and swing phases [3]. Furthermore, a positive correlation has been found between body mass and hip joint quasi-stiffness during the “resilient loading phase” around the terminal stance phase [6]. Similarly, a more recent study emulating reduced gravity showed that hip joint quasi-stiffness increases during pre-swing hip flexion when walking at 0.8 and 1.2 m/s with reduced vertical weight support through a harness. It also confirmed that walking at a faster speed increases hip joint quasi-stiffness during the pre-swing phase [10]. This increased joint quasi-stiffness due to greater added mass or faster walking speed has been observed at other lower-limb joints as well. With as low as 15% body mass added to the upper body of young adults during walking, quasi-stiffness at the ankle joint was found to increase during both the ankle dorsiflexion and plantarflexion phases [11]. A similar effect was observed for older adults who increased their ankle joint quasi-stiffness during late stance when walking faster [5]. It should be noted that age-related muscle loss may influence joint quasi-stiffness in older adults [41].

Overall, quasi-stiffness studies have focused less on the hip than the ankle and knee, and those studies evaluating hip joint quasi-stiffness have been mostly focused on young adults [3,6,10]. As middle-aged or older adults are more likely to be the target population for hip exos, it would be valuable to examine whether such age groups adopt similar hip joint quasi-stiffness relationships as young adults. One prior study compared young and middle-aged adults walking at multiple speeds and found no significant difference between the two age groups in hip/knee/ankle joint quasi-stiffness during the braking phase of stance, despite there existing differences in joint mechanical work [2]. Other prior studies have shown that the amplitude of hip joint mechanical work during walking increased among both young [42] and middle-aged adults under increased walking speeds [2]. Likewise, the amplitude of hip joint mechanical work during walking increased among young adults with added mass, but the result is currently unknown for other age groups [21,42]. Given that exos are likely to alter walking speeds and added mass to the user, reporting on both quasi-stiffness and mechanical work across multiple age groups may help to establish whether clear relationships exist between the two or not. Such information is important as exos have the potential to assist aging adults in achieving higher mobility and thus improve their quality of life and independence [43].

To summarize, knowledge of joint quasi-stiffness during emulated hip exo conditions can improve a person’s mobile performance. Previous studies have characterized hip joint quasi-stiffness during walking between the terminal stance phase and early swing phase [3,6], or across different time windows during the stance phase with reduced gravity [10]. Some have focused on other joints [11,42], and some have focused on a different phase of the gait [2]. However, to our knowledge, the hip joint quasi-stiffness response to changes in walking speeds and with added mass around the pelvis and thighs, emulating a hip exo, has not yet been characterized. To maximize the translational impact of this study, we propose to analyze the sagittal plane hip joint quasi-stiffness during the hip extension phase and the hip flexion phase for both young and middle-aged, healthy adults. To emulate hip exos with different mass amounts and mass distributions, various amounts of added mass were attached to the participants’ lower body (pelvis and both thighs). Specifically, the total added mass, 0 to ~10 kg, was representative of the mass range of most recent exos with hip joint actuation [20,25]. The sagittal plane hip joint mechanical work generated during the two phases was also evaluated to observe whether any relationships could be described between quasi-stiffness and the mechanical energy consumption, or generation, at the hip joint. The two phases analyzed were chosen because, during these gait timings, the magnitude of the hip joint moment is high, and the hip joint has been shown to exhibit “spring-like” properties [3,6,10]. The “spring-like” properties imply that the hip joint moment–angle relation is expected to have relatively linear slopes during these stages, making for good fits of the hip joint quasi-stiffness. Equally important is that these timings are when exos have been designed to store/absorb and release/generate mechanical energy at the hip joint [16].

Based on previous reports [2,3,5,6,9,10,11,21,42], for this study the following hypotheses were tested during the hip extension and hip flexion phases, respectively: (1) when walking at a higher speed, healthy adults would increase their hip joint quasi-stiffness; (2) when wearing hip-exo-like added mass on the pelvis and thighs, healthy adults would increase their hip joint quasi-stiffness. Lastly, comparisons of hip joint quasi-stiffness and hip joint mechanical work between young adults and middle-aged adults were performed to provide a better understanding of how age influences the response to these factors of walking speed and added mass.

## 2. Materials and Methods

### 2.1. Participants

Within this work, 34 adults, who participated in a parallel study [18], provided written consent and were considered to be healthy, with a BMI < ~30 kg/m^2^, normal blood pressure, capable of walking without any assist, and no reported lower limb injuries within 6 months prior to the participation. The study was approved by the Institutional Review Board at the University of Dayton. Among the 34 participants, collected and processed biomechanical data from 5 participants were not included in the final statistical analysis of this study as their inclusion did not meet the linear fit quality for estimating quasi-stiffness; see Section 2.6 quasi-stiffness estimates in the Methods section for more details. Out of the 29 participants analyzed in this study, 13 of them were young adults (age 22.5 ± 3.9 years, body mass 71.5 ± 9.0 kg, body height 1.73 ± 0.07 m, BMI 23.9 ± 3.4 kg/m^2^, 5 were female), and 16 of them were middle-aged adults (age 44.6 ± 8.9 years, body mass 75.4 ± 14.0 kg, body height 1.70 ± 0.09 m, BMI 25.9 ± 3.6 kg/m^2^, 10 were female).

### 2.2. Study Design

The participants performed treadmill walking at three participant-specific speeds, when wearing added mass of different amounts on both the pelvis and the thighs. Each participant started with 2 min of overground free walking, whose average speed, measured using a measuring wheel, was used as the participant-specific treadmill walking speed (100% speed). Within this study, the 100% speed was on average 1.059 ± 0.144 m/s for the young adults, and 1.058 ± 0.115 m/s for the middle-aged adults. Each participant was fitted with retroreflective markers that were attached to the torso, pelvis, thighs, shanks, and feet. A full list of retroreflective markers and a figure showing the placement of all retroreflective markers can be found in the parallel study [18]. Cylindrical tungsten alloy bars each with a mass of 1.8 lb (Midwest Tungsten Service, Willowbrook, IL, USA) were attached to the pelvis and thigh segments. Details about the quantity and locations of tungsten bars are provided in Table 1.

### 2.3. Biomechanical Data Collection

Retroreflective marker trajectories (at 150 Hz) and ground reaction forces (at 1500 Hz) were collected using a 10-camera Vicon motion capture system (Vicon Motion Systems Ltd., Oxford, UK) and Bertec fully instrumented split-belt treadmill (Bertec Corporation, Columbus, OH, USA). A set of baseline data (baseline) with no added mass were collected first. The baseline data consisted of a static trial collection followed by 3 continuous 60 s treadmill walking trial collections; during each period, the treadmill was set to 1 of the 3 randomized speeds: 100%, 115%, and 130% participant-specific speed. After the baseline data, nine different sets of full factorial added mass conditions were collected using the same procedures (Table 1). The sequence of these added mass conditions was also randomized. Although there was typically no break within the same set of conditions, each participant was offered up to five minutes, if needed, of resting time between every two sets of conditions.

### 2.4. Biomechanical Data Pre-Processing

Using Nexus (Vicon Motion Systems Ltd., Oxford, UK), automated marker labeling was applied and a custom gap-filling pipeline was run with manual corrections afterward when mislabeling was found. Both the retroreflective marker trajectories and the ground reaction forces were low-pass filtered using Visual3D (C-Motion Inc., Germantown, MD, USA) with a 2-way 4^th^-order Butterworth filter at 15 Hz. For each participant’s data, a participant-and-condition-specific skeletal model was created using Visual3D. This model featured a torso segment (no hands, arms, or head), a pelvis segment, left and right thigh segments, left and right shank segments, and left and right foot segments. The average vertical ground reaction force captured during the baseline static trials was used to calculate each participant’s body mass. Based on this, the default Visual3D segment mass was used for this baseline mass condition model of the participant and was adjusted accordingly to create the models for each set of added mass conditions to reflect the corresponding segment mass changes caused by the added mass. Gait cycles where a foot crossover occurred (e.g., when a foot contacted the treadmill belt on the opposite side) were recorded and excluded from further analysis, as accurate joint kinetics require clean foot contact. Each participant’s joint kinematic and kinetic metrics were calculated using Visual3D. Specifically, for this study, the left and right sagittal plane hip joint angles, moments, and powers were analyzed. The Visual3D-processed data were then exported to MATLAB (The MathWorks Inc., Natick, MA, USA) for the calculation of the hip joint quasi-stiffness and the hip joint mechanical work, defined in Section 2.6 and Section 2.7 in the Methods section.

### 2.5. Late-Stance Segmenting

During the terminal stance and pre-swing phases, the hip moment–angle relationship moves in a greater extension angle and higher flexion moment direction, before it starts moving in the opposite direction (Figure 1c). For this reason, this region was divided into two phases. The hip extension motion phase started at a 0 radian hip joint angle and ended at the maximum hip extension angle. The hip flexion motion phase started at the maximum hip extension angle and ended at toe-off. These definitions were derived from a previous study [6], and were tailored for this study for feasibility reasons.

### 2.6. Quasi-Stiffness Estimates

Joint quasi-stiffness is not a direct representation of joint stiffness, but is instead a lower-order approximation, as has been demonstrated in a previous comparison [1]. This is because the slope of the moment–angle plot does not take the angular acceleration and velocity into consideration. With that said, in this study, if the moment–angle plot has a constant slope in a region, the hip joint stiffness can be approximated using the slope over this region, as seen in previous studies [6,10].

Figure 1b shows the representative hip joint angle trajectories from 1 participant for a 60 s trial. Here, definitions of the phases are given. The hip extension motion phase starts right after the hip joint angle reaches 0 radian and enters extension. In these representative hip joint angle trajectories, this timing occurred at around 25–35% of the gait cycle. As the horizontal values on the plot were all time-normalized, meaning that all of the horizontal values were scaled by a constant, the relatively constant slope before and after the hip extension motion phase starting point indicates that the participant was swinging the thigh posteriorly at a constant angular speed. However, this speed did drop as it approached the end of the phase with a maximum hip extension angle. Entering the hip flexion motion phase, the angular speed gradually increased and maintained this constant angular speed beyond the end of the phase, defined by toe-off; after which, the leg continued swinging forward until the end of the gait cycle.

The sagittal plane hip joint moment followed a similar pattern; see Figure 1a. At the beginning of the hip extension motion phase, the moment was close to 0 Nm/kg, and it increased at a relatively constant rate until the phase ended with the maximum hip extension angle and maximum hip flexion moment. These positive internal moments created angular acceleration in the hip flexion direction (motion), slowing down the backward swing of the thigh together with the external torque via the gravitational force, and peaking at the end of the phase. Entering the hip flexion motion phase, the decreasing trend of the hip moment was steady until toe-off. During normal walking, throughout both phases, these internal hip joint moments in the sagittal plane continued providing the thigh with torques in the direction of flexion. These moment–angle patterns occurred for all of the participants as they walked on the treadmill, and the constant slopes during both of these two phases were validated via high R-squared values. However, as the slopes are expected to be much higher around the maximum hip extension angle, where the two phases intersect, a slope should not only be taken around this peak point but instead should also include a minimum number of data points [6].

To quantify the hip moment–angle relationship during the hip extension and flexion motion phases, the curve fitting function in MATLAB “fit()” with a model of “poly1” was applied to use a linear polynomial curve to represent the moment–angle relationship. For each gait cycle, the quasi-stiffness for each phase was determined from the slope of the moment–angle relationship. The coefficient of determination, R-squared, and root mean squared error, RMSE, were also calculated for each gait cycle to represent the goodness of fit. As R-squared varied because of the application of a linear curve to represent a portion of the nonlinear moment–angle relationship, gait cycles with an R-squared value < 0.6 were excluded from the statistical analysis. Furthermore, as the number of data points available to fit a curve may not be sufficient during the hip extension motion phase, gait cycles with less than 10 data points in this phase were also excluded from the statistical analysis. This criterion yielded a minimum duration of 0.06 s for the extension phase to be considered for further analysis, which was similar to the “0.05 s” criterion used in a recent quasi-stiffness study [10]. All gait cycles met this criterion for the hip flexion motion phase. Following this procedure, out of the 34 participants that we recruited in the larger study, data from 5 participants were not included in the statistical analysis, as their R-squared values were low, and their moment–angle data usually came with less than 10 data points during the extension phase. Thus, the data presented in this study reflect the participants whose moment–angle relationship during late stance was fairly linear, and those who did not have consistently low hip extension angles during walking.

### 2.7. Mechanical Work Estimates

Sagittal plane hip joint mechanical work is the time integral of the sagittal plane hip joint power. Hip joint power was calculated within Visual3D via an inverse dynamics process. As seen on the hip joint moment–angle plot, it is equivalent to the area under the quasi-stiffness curve. In this study, for each gait cycle, hip joint mechanical work at each phase was estimated by integrating the hip joint power using the MATLAB “trapz()” function. In this study, the sagittal plane hip joint quasi-stiffness equaled the slopes of the moment–angle plot in the hip extension and flexion phases, and the quasi-stiffness described in each stage how rapidly the mechanical work increased or decreased along the change in hip joint angle or moment.

### 2.8. Statistical Analysis

Statistical analysis was performed to determine the influences of age, walking speed, and added mass on hip joint quasi-stiffness and mechanical work. Overall, symmetry was assumed in the healthy participants. Thus, the metrics from both the right and left sides were combined during the statistical analysis. NCSS 2021 (NCSS, LLC, Kaysville, UT, USA) was used to perform the statistical analysis. The 3 factors analyzed were as follows: 1. age category (young or middle-aged), 2. added mass condition, and 3. walking speed level (100%, 115%, and 130%). Three-way repeated measures (within-subject) analysis of variance (ANOVA) was selected as the NCSS procedure during the statistical analysis. F-Tests with Geisser–Greenhouse adjustments were run first, followed by Tukey–Kramer pairwise comparisons. The significance level was set at 0.05 throughout. Metric averages were solved and compared using Microsoft Excel (Microsoft Corporation, Redmond, WA, USA).

## 3. Results

In this section, the hip joint quasi-stiffness, mechanical work values, and their statistical differences are presented with respect to three factors: age category, walking speed, and added mass conditions. Specifically, the following are presented: 1. goodness of fit when obtaining the slopes of the moment–angle plots, 2. ANOVA results, and 3. pairwise comparisons of metric values between different age categories (between-subject), between different walking speed levels (within-subject), and with different added mass conditions (within-subject), as well as the significance of their influence on the metric values. Detailed average metric values of every experimental condition and their significance with respect to the baseline conditions can be found in Appendix A.

### 3.1. Overall Results

An overview of the participant-specific results can be found in Appendix A. These tables show basic information about each participant, along with the analyzed metrics. The metrics include the total number of the hip extension and flexion quasi-stiffness values, the average R-squared and root mean squared error for the curve fitting, the average number of frames for every hip extension and flexion quasi-stiffness value, the range of hip extension and flexion quasi-stiffness, and the average mechanical work. Furthermore, also included are the metric averages and standard deviations across all of the participants. To summarize, across all of the participants, quasi-stiffness during hip extension (K_EX_) was 2.651 ± 0.675 (SD) Nm/kg/rad, and during hip flexion (K_FL_) was 2.999 ± 1.083 (SD) Nm/kg/rad. Across all of the participants, mechanical work during hip extension (W_EX_) was −0.082 ± 0.063 (SD) J/kg, and during hip flexion (W_FL_) was 0.062 ± 0.031 (SD) J/kg. These results were achieved with a reasonable goodness of fit. For hip extension, there were on average 36 frames of data available for the curve fitting process, and an average R-squared value of 0.93 was achieved over 2820 gait cycles for each participant. Similarly, with 20 frames of data, fitting for hip flexion achieved an average R-squared value of 0.90 across 2884 gait cycles for each participant.

### 3.2. Goodness of Fit

The goodness of fit, represented by R-squared and RMSE, is shown in Appendix A. R-squared and RMSE, though with some variations, maintained their values across all of the participants. Out of the analyzed 169,677 hip joint quasi-stiffness values (including both the left and right hip joints during both hip extension and flexion), 266 had an R-squared value < 0.6 and were excluded from statistical analysis, 158,590 had an R-squared value ≥ 0.8, and 116,663 had an R-squared value ≥ 0.9.

### 3.3. F-Test

F-Tests with Geisser–Greenhouse adjustments for each metric showed the following: the added mass condition (individual added mass combination and individual walking speed level) was a significant factor during both extension and flexion: *p* < 0.0001 with power > 0.9999; the walking speed level itself was a significant factor during extension: *p* < 0.0001 with power > 0.9999; and age category was not a significant factor.

### 3.4. Pairwise Comparisons

#### 3.4.1. Age Category

For the age factor, the metric means and their *p*-values are presented in Table 2. Most importantly, the average hip flexion quasi-stiffness (K_FL_) for the middle-aged adults was significantly higher by 27.4% from that of the young adults (*p* = 0.0036). However, the hip extension quasi-stiffness (K_EX_) and hip joint mechanical work values during both the hip extension and flexion stages (W_EX_ and W_FL_) for the middle-aged adults were not significantly different compared to the young adults.

#### 3.4.2. Walking Speed

For the walking speed factor, the metric means and their *p*-values are presented in Table 3. For both young and middle-aged adults, increases in walking speed consistently increased the hip extension and flexion quasi-stiffness (K_EX_ and K_FL_) values (all *p* < 0.001). Stiffness (K_EX_ and K_FL_) increases between 8.6% and 34.1% with respect to the baseline (100%) speed were observed. Work values (W_EX_ and W_FL_) also increased in magnitude, between 16.5% and 59.0%, with respect to the baseline (100%) speed. The relative increases in both quasi-stiffness and work values among the young adults were higher than those of the middle-aged adults.

#### 3.4.3. Added Mass Condition

Lastly, the influence of added mass conditions is presented. Given the full factorial combination of these mass conditions, their relative differences from the baseline conditions are shown in such a format that will directly represent their relative mass location in space. See Figure 2 and Figure 3 for quasi-stiffness during hip extension and flexion and Appendix A for joint mechanical work during hip extension and flexion. Within each small cluster on these figures, there is a baseline metric value on the left, and nine relative differences from this baseline metric value for the nine added mass conditions. Within each cluster, conditions sitting higher have a center of added mass closer to the pelvis, and those sitting lower have a center of added mass closer to the knee joint. Within each cluster, conditions with higher amounts of total added mass are horizontally farther away from the baseline conditions. In short, the higher a condition sits within its cluster, the closer its center of added mass is to the pelvis, and the further right a condition sits within its cluster, the greater its amount of added mass. Figure 4 and Figure 5 show the changes in the sagittal plane hip joint moment–angle fit, i.e., quasi-stiffness estimate during extension and flexion.

With added mass, hip extension quasi-stiffness, K_EX_, generally increased for young adults (Figure 2 and Figure 4). The maximum relative changes in hip extension quasi-stiffness, K_EX_, for the young adults were observed with 21.6 lb of added mass (mass condition: “33”), which resulted in an increase of 11.6% from the baseline. With respect to K_EX_ of the middle-aged adults, no trending pattern in quasi-stiffness change was observed. Across the 3 levels of walking speed, the maximum relative changes observed were with 10.8 lb of added mass (mass conditions “12” and “21”), which resulted in a decrease of 4.8% from the baseline. This was followed by the heaviest added mass condition, “33”, which increased K_EX_ by 4.6% with respect to the baseline. For the young adults, changes in K_EX_ were statistically significant for 26 out of 27 comparisons. The exception was for 7.2 lb of added mass (mass condition: “11”) at the 130% speed. Typically, the highest changes were caused by mass conditions with a higher amount of added mass. For the middle-aged adults, however, only 14 out of 27 of the changes were statistically significant. Out of the significant items, the changes were in different directions, and mass conditions with a higher amount of added mass did not necessarily induce greater changes.

With added mass, hip flexion quasi-stiffness, K_FL_, saw a decrease from its baseline (Figure 3 and Figure 5). For the young adults, the maximum relative change was observed with 18.0 lb of added mass (mass condition: “32”), which reduced K_FL_ by 16.5% from the baseline. For the middle-aged adults, the maximum relative change was observed with 14.4 lb of added mass (mass condition: “22”), which reduced K_FL_ by 4.8%. For the young adults, 27 out of 27 changes were statistically significant. Typically, a heavier added mass resulted in a greater reduction in K_FL_ from the baseline. For the middle-aged adults, however, only 14 out of 27 of the changes were statistically significant. Unlike the young adults, changes among the middle-aged adults did not appear to follow a trend. The heaviest added mass did not induce greater changes in K_FL_.

Mechanical work at the hip over the two stages increased with added mass (Appendix A). For both the young and middle-aged adults, the changes in W_EX_ and W_FL_ were all statistically significant. For the young adults, 14.4 lb of added mass increased work during hip extension, W_EX_, by 30.2% with respect to the baseline. For the middle-aged adults, 14.4 lb of added mass (mass condition: “31”) increased W_EX_ by 36.2% with respect to the baseline. For the young adults, 18.0 lb of added mass (mass condition: “32”) increased work during hip flexion, W_FL_, by 38.5% with respect to the baseline. For the middle-aged adults, 14.4 lb of added mass (mass condition: “31”) increased W_FL_ by 21.7% with respect to the baseline. Mass conditions with a high center of added mass typically yield the greatest changes. This trend was consistent across both the young and middle-aged adults and for the work during both hip extension and flexion (W_EX_ and W_FL_). Detailed metric values are shown in Appendix A.

## 4. Discussion

### 4.1. Hip Joint Quasi-Stiffness

The goal of this study was to investigate the influence of the combination of increased walking speeds and added mass on hip joint quasi-stiffness among healthy adults. Based on the results of this study and prior studies [3,10], the hypothesis that increasing the walking speed will increase quasi-stiffness appears to be validated for both young and middle-aged adults. However, the hypothesis that quasi-stiffness increases with emulated hip exo added mass is more complex. In this study, we found that young adults support this hypothesis during hip extension, but the opposite, a reduction from baseline, occurs for hip flexion. Middle-aged adults showed mixed results for both hip extension and flexion changes that did not support such a generalization but instead required knowledge of the added mass amounts and locations on the body.

It is worth noting, before going into a discussion regarding the exact details of the influence of walking speeds and added mass on quasi-stiffness, that the approximations of such metrics using linearization were validated as being acceptable for both hip extension and flexion; see Appendix A.

### 4.2. Comparisons between Young and Middle-Aged Adults

Considering all of the experimental conditions during extension, young adults did not exhibit a different quasi-stiffness from that of middle-aged adults; see Table 2. This lack of significant differences in quasi-stiffness between the two age groups during extension has been observed previously over the braking phase as well [2]. A similar observation was made for the mechanical work during hip extension, though with a greater relative difference (9.5% for W_EX_ vs. 0.8% for K_EX_) between the two age groups.

During hip flexion, a statistical difference between quasi-stiffness in young and middle-aged adults was observed; see Table 2. Middle-aged adults’ quasi-stiffness was 27.4% higher than that of young adults. The moment–angle diagrams show that the difference in quasi-stiffness can be attributed to differences in the hip angle at the end point (toe-off) between the two age groups (Figure 5). No statistical differences were observed between the two groups in work during hip flexion when mechanical work was generated at the hip joint. Given the statistical differences observed between the two groups, the following discussion will be split into the influences of walking speed and added mass within each age group.

### 4.3. Effects of Walking Speed on Young and Middle-Aged Adults

For both young and middle-aged adults, walking at a higher speed increased hip joint quasi-stiffness. Thus, the hypothesis that walking faster increases quasi-stiffness was further validated in this study within both groups. Specifically, increased walking speed significantly and consistently altered the quasi-stiffness during both extension and flexion; see Table 3. The changes were such that a 30% increase in walking speed increased the mechanical work by almost 60% during hip flexion for young adults. This significant impact from increased walking speed aligns with what has been reported at the ankle joint by a previous study [5].

The speed-induced quasi-stiffness increases observed in this study are comparable to those reported in previous studies with predominantly healthy, young adults [6,10]. To be specific, this study had both young and middle-aged participants walking on average at 1.058–1.375 m/s (100% and 130% speeds), which resulted in a quasi-stiffness range of 0.873–6.437 Nm/kg/rad during hip extension, and 0.560–9.157 Nm/kg/rad during hip flexion. Compared to the literature, a prior study assessed quasi-stiffness at walking speeds from 0.75 m/s to 2.6 m/s and also reported higher mean quasi-stiffness with faster walking speeds [6]. The quasi-stiffness range for that study yielded 1.029–23.386 Nm/kg/rad for hip extension and 0.145–16.336 Nm/kg/rad for hip flexion. Additionally, a different prior study reported hip flexion quasi-stiffness to increase from 2.750 Nm/kg/rad at a walking speed of 0.8 m/s to 4.354 Nm/kg/rad at 1.2 m/s [10]. Thus, this study expands upon what is known in the literature on the effect of walking speed on hip quasi-stiffness, as it shows that such an effect is consistent across both young and middle-aged groups.

### 4.4. Effects of Added Mass on Young Adults

The influence of added mass on quasi-stiffness is consistent for young adults: increases during hip extension and decreases during hip flexion. Adding mass to the lower body increased quasi-stiffness and kinetic energy absorption during hip extension (~30–55% of the gait cycle) and decreased quasi-stiffness with a higher kinetic energy generation during hip flexion (~55–65% of the gait cycle). However, increasing the amount of added mass does not always amplify those influences. When more added mass is transferred from the pelvis to the thighs, the quasi-stiffness mostly decreases during extension, and increases during flexion. This transfer of added mass from the pelvis to the thighs also reduces kinetic energy absorption during extension and kinetic energy generation during flexion.

The hypothesis that added mass increases quasi-stiffness is validated during hip extension for young adults. The changes are almost all statistically significant; see Figure 2. When relating to the moment–angle diagrams, the starting point is nominally the same but the end point (at the maximum extension angle) is often increased more by the moment than the angle; see Figure 4. Analyzing this further, during extension, increased quasi-stiffness will absorb more kinetic energy over a small angular change. An increased rate of kinetic energy absorption leads to increased kinetic energy absorption at the hip during extension; see Appendix A.

Both the amount and the distribution of added mass play a role in quasi-stiffness for young adults. During hip extension, when added mass is evenly distributed across the pelvis and thighs (mass conditions: “11”, “22”, and “33”), the higher the amount of added mass, the greater its influence on quasi-stiffness. With total added mass of <14.4 lb, young adults steadily increased their quasi-stiffness when more added mass was added unevenly across the pelvis and thighs (mass conditions: “31”, “22”, and “13”). However, the patterns after this were not consistent. The potential reasoning for this can be that increased quasi-stiffness during extension has a negative impact from a kinetic energy point of view, as it would absorb more kinetic energy over a shorter angular change.

Given that young adults reduced their quasi-stiffness with added mass during hip flexion, see Figure 3, the hypothesis that added mass increases quasi-stiffness is not supported. When relating to the moment–angle diagrams, the observed reduction in quasi-stiffness can be best attributed to the fact that the starting point has less changes than the angular shift towards flexion at the end point, toe-off; see Figure 5. During this stage of returning kinetic energy, demonstrated via positive mechanical work, see Appendix A, with reduced quasi-stiffness, the hip joint generates greater mechanical work over the same angular change.

During hip flexion, the presumption that heavier added mass would result in larger changes was not fully supported. An added mass as little as 7.2 lb (mass condition: “11”) reduced the young adults’ quasi-stiffness by 8.8–13.1% compared to the baseline, whereas a larger added mass (21.6 lb) reduced it by 6.7–15.4%, indicating that reducing the mass added from 21.6 lb to 7.2 lb would not proportionally reduce its influence on quasi-stiffness during flexion.

Although there does not seem to be a one-size-fits-all observation for the influence of the amount of added mass on young adults, some added mass distributions may be more ideal than others. For example, when the center of added mass was lower (closer to the thighs), we observed smaller increases in quasi-stiffness during extension and smaller reductions in quasi-stiffness during flexion, as demonstrated in Figure 2 and Figure 3. A smaller increase in extension quasi-stiffness, see Figure 2, shows that the mechanical work is distributed more evenly over a larger angular change, whereas a smaller reduction in flexion quasi-stiffness, see Figure 3, shows that the mechanical work is distributed less evenly, and more mechanical work will be generated at the beginning of hip flexion. A lower center of added mass means that equal or more tungsten bars were added to the thighs than the pelvis. In Figure 2 and Figure 3 and Appendix A, this falls in the lower half of each cluster (mass conditions: “11”, “12”, “13”, “22”, “23”, and “33”).

One possible reason leading to increased quasi-stiffness during extension is that a higher center of added mass also places more of the added mass closer to the center of mass of the body. During hip extension, the center of mass of the body has just moved in front of the leg and keeps moving forward, while the thigh swings backward, dragging the center of body mass. Since more added mass is placed closer to the center of body mass, the hip joint must increase its quasi-stiffness to slow down the center of body mass. The turning point is when the plantarflexion moment at the ankle joint provides the propulsive force at the center of body mass through the hip joint, while the thigh swings forward. However, reduced hip joint quasi-stiffness during flexion indicates that young adults experienced more difficulty swinging the thigh forward during this phase.

### 4.5. Effects of Added Mass on Middle-Aged Adults

In response to added mass, middle-aged adults performed very differently in comparison to the young adults. Specifically, (1) when walking at the same speeds, middle-aged adults had higher quasi-stiffness during both hip extension (~35–55% of the gait cycle) and flexion (~55–65% of the gait cycle); (2) middle-aged adults had smaller relative changes in their quasi-stiffness when wearing added mass on the lower limb; and (3) middle-aged adults showed higher variability in quasi-stiffness and mechanical work. A smaller number of significant quasi-stiffness changes in response to added mass indicates that middle-aged adults do not systematically alter their quasi-stiffness. However, given the higher variability, predicting quasi-stiffness changes for middle-aged exo users may be difficult. Additionally, variability in quasi-stiffness reduced with most added mass combinations, indicating that although it can be more difficult to predict the quasi-stiffness changes for middle-aged adults, added mass of a hip exo may make such predictions easier by narrowing the range down.

Our results support our hypothesis that added mass increases hip joint quasi-stiffness for middle-aged adults during extension. However, in contrast to the young adults, these changes are smaller, and only apply to limited added mass conditions. In comparison to an added 7.2 lb (mass condition: “11”) where both age groups retained their quasi-stiffness compared to the baseline, the largest changes were caused by 10.8 lb (mass conditions: “12” and “21”). More specifically, these changes were 3.9–4.8%, and even heavier mass conditions did not result in a larger change. For middle-aged adults, the mass condition “12” was unique in that a quasi-stiffness reduction was observed during extension across all speeds. During flexion, however, middle-aged adults did alter their quasi-stiffness more with added mass of higher amounts, but only when the total added mass was no more than 14.4 lb and the center of added mass was not closer to the thighs. As observed in young adults, this should be beneficial for middle-aged adults to retain their kinetic energy at the hip joints by absorbing less while returning less.

Middle-aged adults experienced higher hip joint quasi-stiffness. When comparing the detailed quasi-stiffness values, see Appendix A, the relative changes in the quasi-stiffness were different between the two age groups. In response to added mass, although at all walking speeds both groups increased the quasi-stiffness during extension and decreased it during flexion, the ranges were different. During extension, middle-aged adults started with higher quasi-stiffness at baseline. Walking at 100% speed, they had an average quasi-stiffness of 2.40 Nm/kg/rad, which was increased to 2.51 Nm/kg/rad at most with added mass. However, for young adults walking at the 100% speed, they started with a lower quasi-stiffness (2.24 Nm/kg/rad) and were able to increase their quasi-stiffness to a similar maximum value (2.50 Nm/kg/rad) with added mass. The same trends were observed at both the 115% and 130% speeds. For example, at 130% speed, young adults started at 2.75 Nm/kg/rad, and managed to increase the quasi-stiffness to a maximum of 2.99 Nm/kg/rad, whereas middle-aged adults started higher, at 2.83 Nm/kg/rad, and only peaked at 2.92 Nm/kg/rad.

Similar behavior happened during hip flexion as well. Both young and middle-aged adults reduced their quasi-stiffness due to added mass. Under the most extreme case, when walking at 130% speed, young adults started at 3.40 Nm/kg/rad, and reduced to a minimum of 2.84 Nm/kg/rad, whereas middle-aged adults started with 3.78 Nm/kg/rad, and only managed to reduce their quasi-stiffness to 3.61 Nm/kg/rad. This was a 0.56 Nm/kg/rad reduction for young adults, and only a 0.17 Nm/kg/rad reduction for middle-aged adults. Furthermore, middle-aged adults consistently showed higher standard deviations, with condition standard deviations ranging from 0.93 to 1.33 Nm/kg/rad, as opposed to young adults’ 0.58 to 1.07 Nm/kg/rad; see Appendix A. Their hip joint mechanical work during hip extension had a standard deviation of 0.7 to 0.8 W/kg across all of the conditions, as opposed to young adults’ 0.03 to 0.05 W/kg; see Appendix A. This indicates that middle-aged adults had higher quasi-stiffness and mechanical work variabilities during both extension and flexion.

### 4.6. Implications to Hip Exo Control

In this work, hip joint quasi-stiffness serves as an approximation of hip joint stiffness. For a hip exo with cooperative impedance control, to calculate the required torque output, the exo needs to first acquire an estimate of the wearer’s joint stiffness, which in this case was the hip joint quasi-stiffness. Here, it is worth noting that all quasi-stiffness quantifications discussed in this study were in the sagittal plane only, and so results will not directly translate to other planes such as the frontal plane. With that said, in this study, both young and middle-aged adults significantly increased both extension and flexion quasi-stiffness at faster walking speeds. This suggests that a hip exo should increase its stiffness support during the extension stage to help to absorb the kinetic energy. Similarly, in the flexion stage, a hip exo should increase its joint stiffness to provide hip flexion assistance in order to propel the center of body mass.

With hip-exo-like added mass, responses of young and middle-aged adults differed depending on mass amounts and distributions. Young adults had a majority of the mass conditions yield significant changes and so were generally predictable in their quasi-stiffness response to a hip exo’s added mass: during extension, there was larger quasi-stiffness with heavier added mass and a higher center of added mass; during flexion, there was smaller quasi-stiffness with added mass but not in any generalizable pattern with mass amounts or distributions. In contrast, middle-aged adults had approximately half of the mass conditions yield significant changes with respect to quasi-stiffness and so were much less generalizable. This leads to hip exo designers being able to use this information to intentionally design for the added mass amounts and locations on the body to target this age group. For example, designers could take advantage of the findings that there exist several added mass combinations which will result in a minimal shift of the wearer’s quasi-stiffness, at least under certain walking speeds, and furthermore, these combinations are not necessarily the lightest. With that being said, such mass combinations could still induce a different hip joint angle profile, as the workload is different because of the added mass, and so control algorithms may need to incorporate this information as well. Highlighting another example, hip exo designers could use the findings that since middle-aged adults have a smaller range of quasi-stiffness, estimation errors in a hip exo would be lower for middle-aged adults than for young adults, even if the hip exo control does not make the correct hip joint quasi-stiffness estimate. Thus, this study highlights that designing mass distributions for hip exos based only on information from young adults is not necessarily translational to older populations, even those as close in age as the middle-aged.

## 5. Conclusions

This study supports that both walking speed and added mass on the pelvis and thighs change the sagittal plane hip joint quasi-stiffness during hip extension and flexion among both young and middle-aged adults. The changes were more consistent with walking speed for both age groups, in that they increased their quasi-stiffness during both extension and flexion with higher speeds. The mechanical work also increased with faster walking speeds for both age groups, but by different relative amounts in comparison to quasi-stiffness.

In response to added mass, only extension quasi-stiffness was found to have similarities between the two age groups. Specifically, for both age groups, extension quasi-stiffness remained the same or increased when either the distribution on the pelvis and thighs was even or was biased to the pelvis. However, with added mass biased to the thighs, middle-aged adults had extension quasi-stiffness stay the same or decrease, while for young adults it increased. For flexion quasi-stiffness, young adults exhibited a decrease with added mass but not in any generalizable pattern with respect to the mass amounts or distributions. Conversely, middle-aged adults were found to have flexion quasi-stiffness remain the same or decrease with either an even distribution on the pelvis and thighs or biased to the pelvis, while no change occurred if biased to the thighs. While changes in quasi-stiffness did not always follow the same patterns with added mass amounts or distribution, mechanical work increased for both young and middle-aged adults during hip extension and flexion. Overall, middle-aged adults maintained a higher quasi-stiffness than young adults for all of the conditions but with a smaller range, indicating that their motor control output was elevated but did not tend to vary.

Given that the added mass in this study reflected the mass of a hip exo, we expect similar responses to be observed when a person wears a hip exo. Furthermore, this study found that middle-aged adults showed different added mass adaptations with respect to young adults. In conclusion, this study emphasizes the importance of studying the intended user populations, as parameters for age-specialized hip exos are likely different for achieving effective cooperative control design.

## Figures and Tables

**Figure 1 sensors-23-04517-f001:**
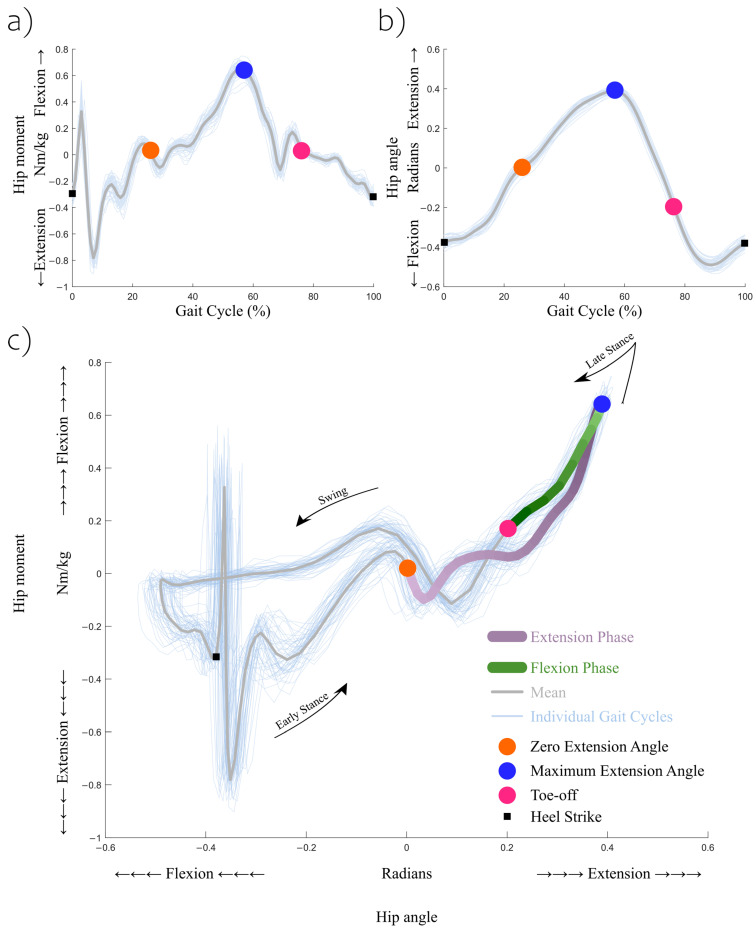
Sagittal plane (**a**) hip joint moment, (**b**) hip joint angle, and (**c**) hip joint moment versus angle from a representative participant during a baseline comfortable speed (100% speed) trial. The hip extension motion phase started at a 0 radian (0°) hip extension angle and ended at the maximum hip extension angle. The hip flexion motion phase started at the maximum hip extension angle and ended at toe-off. Lines were shown darker as each stage proceeded. Events of heel strike, zero extension angle, maximum extension angle, and toe-off were approximate. Quasi-stiffness was estimated for the hip extension and hip flexion phases within each gait cycle.

**Figure 2 sensors-23-04517-f002:**
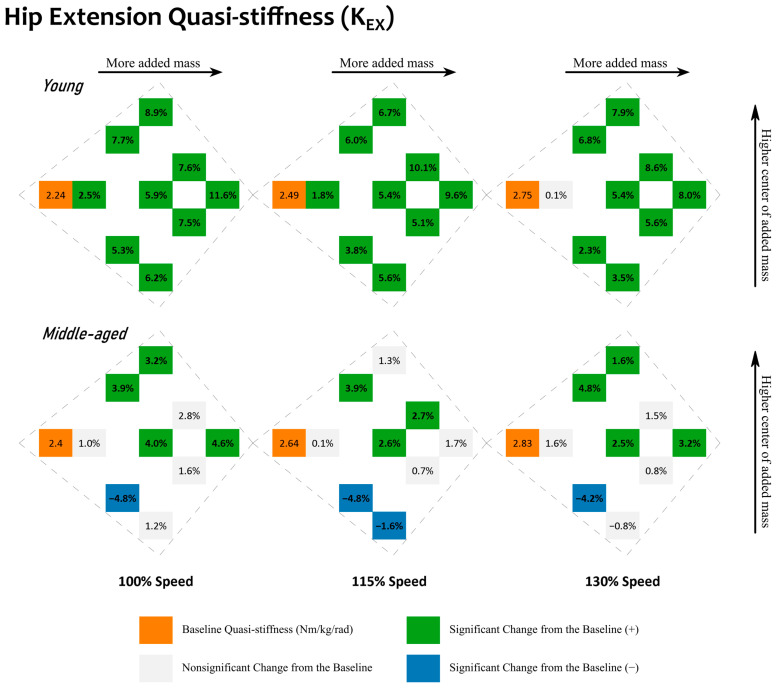
Changes in the sagittal plane hip joint quasi-stiffness during hip extension, K_EX_. Results for young adults are located in the first row. Results for middle-aged adults are located in the second row. Results for walking at different speeds (100, 115, and 130%) are located in the columns. Baseline conditions are highlighted in orange with a unit of Nm/kg/rad. Other values are relative differences with respect to the baseline value within the same cluster. Comparisons that are statistically significant (*p* < 0.05) are in bold with a green background for a positive (+) change and a blue background for a negative (−) change. Comparisons that are nonsignificant have a gray background. Within each cluster, added mass conditions with a higher total amount of added mass sit further to the right, and those with a higher center of added mass sit further away from the bottom.

**Figure 3 sensors-23-04517-f003:**
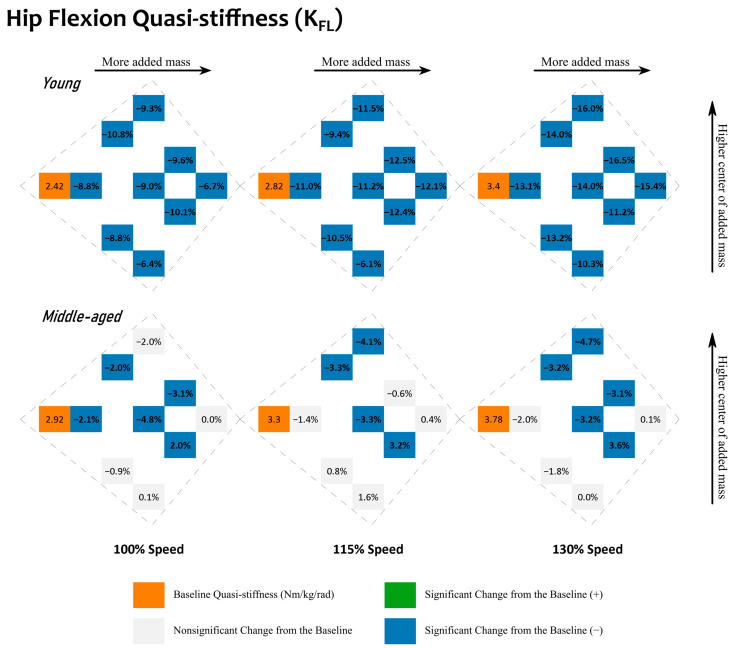
Changes in the sagittal plane hip joint quasi-stiffness during hip flexion, K_FL_. Results for young adults are located in the first row. Results for middle-aged adults are located in the second row. Results for walking at different speeds (100, 115, and 130%) are located in the columns. Baseline conditions are highlighted in orange with a unit of Nm/kg/rad. Other values are relative differences with respect to the baseline value within the same cluster. Comparisons that are statistically significant (*p* < 0.05) are in bold with a green background for a positive (+) change and a blue background for a negative (−) change. Comparisons that are nonsignificant have a gray background. Within each cluster, added mass conditions with a higher total amount of added mass sit further to the right, and those with a higher center of added mass sit further away from the bottom.

**Figure 4 sensors-23-04517-f004:**
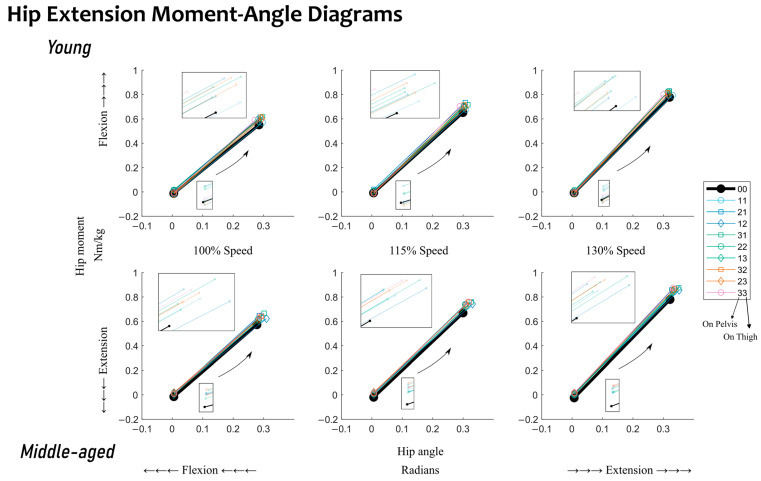
Changes in the sagittal plane hip joint moment–angle fitting, i.e., quasi-stiffness estimate during hip extension, K_EX_. Results for young adults are located in the first row. Results for middle-aged adults are located in the second row. Results for walking at different speeds (100, 115, and 130%) are located in the columns. Baseline conditions, i.e., walking without any added mass, are highlighted in black. The legend follows the same added mass conditions as those of Table 1. Conditions with even distributions of added mass on the pelvis and thigh are indicated with a circle. Conditions with more added mass on the pelvis than the thigh are indicated with a square. Conditions with less added mass on the pelvis than the thigh are indicated with a diamond. Conditions with the same amount of total added mass are indicated with the same line color. Inset boxes show zoomed-in views of the endpoints of the linear fit.

**Figure 5 sensors-23-04517-f005:**
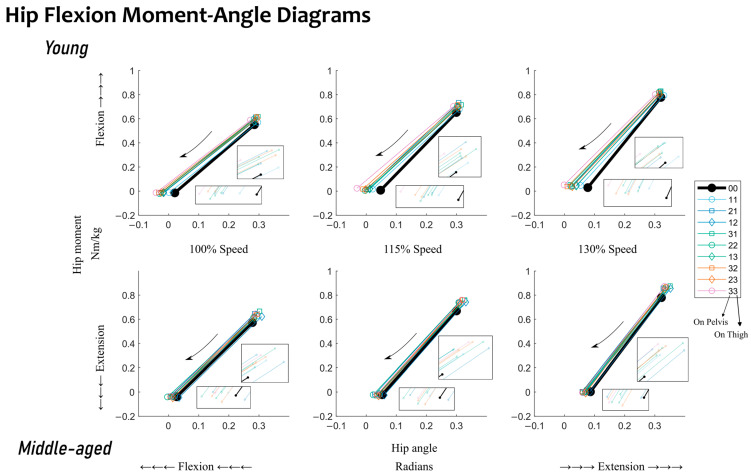
Changes in the sagittal plane hip joint moment–angle fitting, i.e., quasi-stiffness estimate during hip flexion, K_FL_. Results for young adults are located in the first row. Results for middle-aged adults are located in the second row. Results for walking at different speeds (100, 115, and 130%) are located in the columns. Baseline conditions, i.e., walking without any added mass, are highlighted in black. The legend follows the same added mass conditions as those of Table 1. Conditions with even distributions of added mass on the pelvis and thigh are indicated with a circle. Conditions with more added mass on the pelvis than the thigh are indicated with a square. Conditions with less added mass on the pelvis than the thigh are indicated with a diamond. Conditions with the same amount of total added mass are indicated with the same line color. Inset boxes show zoomed-in views of the endpoints of the linear fit.

**Table 1 sensors-23-04517-t001:** Experimental conditions with amount of tungsten bars on each segment.

Added Mass Condition	Tungsten Bar Quantity on Each Segment				Total Added Mass lb (kg)
	Pelvis (Left Side)	Left Thigh	Pelvis (Right Side)	Right Thigh	
00 (baseline)	0	0	0	0	0.0 (0.00)
11	1	1	1	1	7.2 (3.27)
12	1	2	1	2	10.8 (4.90)
13	1	3	1	3	14.4 (6.53)
21	2	1	2	1	10.8 (4.90)
22	2	2	2	2	14.4 (6.53)
23	2	3	2	3	18.0 (8.16)
31	3	1	3	1	14.4 (6.53)
32	3	2	3	2	18.0 (8.16)
33	3	3	3	3	21.6 (9.80)

On the pelvis, the tungsten bars were located along the left and right sides symmetrically. On each thigh, the tungsten bars were at relatively the same height along the lateral circumference (2 bars on the side, and 1 in the front, where on each thigh the tungsten bar in the middle of the 3 was present only for the mass conditions with an odd number of tungsten bars on each thigh).

**Table 2 sensors-23-04517-t002:** Comparisons between age categories.

	Young	Middle-Aged	Difference (*p*-Value)
K_EX_ (Nm/kg/rad)	2.639	2.660	0.8% (0.9179)
K_FL_ (Nm/kg/rad)	2.598	3.310	**27.4% (0.0333)**
W_EX_ (J/kg)	−0.078	−0.086	9.5% (0.7457)
W_FL_ (J/kg)	0.059	0.064	7.8% (0.6392)

Increases are all with respect to the young adults; bold item: statistically different with *p* < 0.05. K_EX_ and K_FL_ are, respectively, the sagittal plane hip joint quasi-stiffness during extension and flexion; W_EX_ and W_FL_ are, respectively, the sagittal plane hip joint mechanical work during extension and flexion.

**Table 3 sensors-23-04517-t003:** Comparisons between walking speeds.

	Speed	K_EX_ (Nm/kg/rad)	K_FL_ (Nm/kg/rad)	W_EX_ (J/kg)	W_FL_ (J/kg)
Young	100%	2.382	2.225	−0.065	0.046
	115%	**2.623 (10.1%)**	**2.547 (14.4%)**	**−0.078 (19.1%)**	**0.059 (30.1%)**
	130%	**2.887 (21.2%)**	**2.985 (34.1%)**	**−0.090 (38.3%)**	**0.072 (59.0%)**
Middle-aged	100%	2.443	2.880	−0.073	0.051
	115%	**2.653 (8.6%)**	**3.277 (13.8%)**	**−0.085 (16.5%)**	**0.064 (24.1%)**
	130%	**2.864 (17.2%)**	**3.731 (29.6%)**	**−0.098 (34.4%)**	**0.076 (49.2%)**

Values in the parentheses are relative differences from the 100% speed condition within the same age category; bold items: statistically different with respect to the 100% speed condition with *p* < 0.001. K_EX_ and K_FL_ are, respectively, the sagittal plane hip joint quasi-stiffness during extension and flexion; W_EX_ and W_FL_ are, respectively, the sagittal plane hip joint mechanical work during extension and flexion.

## Data Availability

Not applicable.

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
