# Peer review of "Effects of Walking Speed and Added Mass on Hip Joint Quasi-Stiffness in Healthy Young and Middle-Aged Adults"

_sensors, 2023, doi:10.3390/s23094517_

Round 1

Reviewer 1 Report

The study presents interest insights on hip quasi-stiffness at 3 walking speed levels and 9 added mass distributions.

The graphic presentation is particularly clear and well presenting, allowing for immediate visualization of the results.

No particular aspects should be revised.

Reviewer 2 Report

The research methodology follows the stated theme of the article. The results, discussion and conclusions refer to the authors' statement: effects of walking speed and added mass on hip joint quasi-stiffness in healthy young and middle-aged adults.

The authors have focused on two categories of adults, young and middle-aged, selected according to some criteria.
I suggest paying attention to the following aspects:
1. Do the subjects have almost the same professional activity, habits and lifestyle? Is their medical history almost identical?

2. Do the authors consider collaborating with experts from the medical field on the chosen topic? I think they can make valuable suggestions on prevention and medical care.

3. What further research can be done on this topic? Do the authors consider investigating how walking speed and added mass affect quasi-stiffness of the hip joint as the person gets older (same selection)?

Reviewer 3 Report

A study was conducted to investigate the quasi-stiffness of the hip joint and its impact on walking speed with added mass. While it is known that walking speed can be affected by both joint stiffness and muscle activity, this study focused specifically on the quasi-stiffness of the hip joint. I think it would be better to also mention the impact of muscle loss on walking speed when adding mass.

Evans, W. J., & Lexell, J. (1995). Human aging, muscle mass, and fiber type composition. The Journals of Gerontology Series A: Biological Sciences and Medical Sciences50(Special_Issue), 11-16.

Farley, C. T., & Ferris, D. P. (1998). 10 biomechanics of walking and running: Center of mass movements to muscle action. Exercise and sport sciences reviews26(1), 253-286.

Liu, M. Q., Anderson, F. C., Schwartz, M. H., & Delp, S. L. (2008). Muscle contributions to support and progression over a range of walking speeds. Journal of biomechanics, 41(15), 3243-3252.

Please add the validity of the sample size of the test.

Reviewer 4 Report

This is a typical data paper on quasi-stiffness of human hip which may serve the developpement of  exoskeleton's. 

The paper presents the hip quasi-stiff-ness at 3 walking speeds. Gathered data concern 13 young and 16 middle-aged 10 adults.

The Data gathering processing need to be well detailed, the experimental gathering test bench should be presented in a generic manner. Adding a figure explaining the marking processing, the markers prositions and also the detailed process of manual corrections of paragraph 2.1. 

Please clearly indicate what are the key processing phase of the data gathering in a flow chart at the level of paragraph 2.

Please add the theorical backgound of the added masses placement, those masses are supposed to represnet an exo mass, so we need to understand how the placement of those masses where motivated, and how is it argued. 

More technichal details are needed about the sensibilities and resolution of the used tracking system should be added. 

Details about how the mechanical work was estimated are needed, authors just cited a matlab function, instead of explaining how the mech work is derived using their data... The matlab function is just an implementation .... 

Please include and discuss data from table S1 in the paper,  to help reader. 

The study focuses on the sagittal plan analysis, while recent biomechanical analysis combines both sagittal and frontal plans 

The Data presented in the supplementary files are intersting for biomechanical engineering and may interests robotics community as well. 

Round 2

Reviewer 3 Report

I have verified that this paper has been properly revised.

Reviewer 4 Report

Authors comments in paper are minor, while their responses to my comments are positive, I would recommend to publish this arranged version of the manuscript.